# Local and Contralateral Effects after the Application of Neuromuscular Electrostimulation in Lower Limbs

**DOI:** 10.3390/ijerph17239028

**Published:** 2020-12-03

**Authors:** Elisa Benito-Martínez, Diego Senovilla-Herguedas, Julio César de la Torre-Montero, María Jesús Martínez-Beltrán, María Mercedes Reguera-García, Beatriz Alonso-Cortés

**Affiliations:** 1San Juan de Dios School of Nursing and Physical Therapy, Comillas Pontifical University, 28350 Madrid, Spain; elisabenito@comillas.edu (E.B.-M.); diegosenovilla@hotmail.com (D.S.-H.); juliodelatorre@comillas.edu (J.C.d.l.T.-M.); mjesus.martinez@comillas.edu (M.J.M.-B.); 2SALBIS Research Group, Faculty of Health Sciences, Campus of Ponferrada, University of León, 24401 Ponferrada, Spain; mmregg@unileon.es

**Keywords:** thermography, electrostimulation, cross-education, rehabilitation

## Abstract

Neuromuscular electrostimulation (NMES) has been used mainly as a method to promote muscle strength, but its effects on improving blood flow are less well known. The aim of this study is to deepen the knowledge about the local and contralateral effects of the application of symmetric biphasic square currents on skin temperature (Tsk). An experimental pilot study was developed with a single study group consisting of 45 healthy subjects. Thermographic evaluations were recorded following the application of NMES to the anterior region of the thigh. The results showed an increase in the maximal Tsk of 0.67% in the anterior region of the thigh where the NMES was applied (*p* < 0.001) and an increase of 0.54% (*p* < 0.01) due to cross-education effects, which was higher when the NMES was applied on the dominant side (0.79%; *p* < 0.01). The duration of the effect was 20 min in the dominant leg and 10 min in the nondominant one. The application of a symmetrical biphasic current (8 Hz and 400 μs) creates an increase in the maximal Tsk at the local level. A temperature cross-education effect is produced, which is greater when the NMES is applied on the dominant side. This could be a useful noninvasive measurement tool in NMES treatments.

## 1. Introduction

It has been repeatedly reported that neuromuscular electrostimulation (NMES) is a useful tool and gold standard in several rehabilitation procedures [1,2]. Among its most well known benefits are the improvements of muscle strength [3] and neuro-muscular activation [4]. However, some studies have shown increases in blood flow after the application of low-frequency symmetric biphasic square currents [3,5] or diadynamic currents [6]. Other vascular effects have been demonstrated using neuromuscular electrostimulation (NMES) by applying frequencies close to 8 Hz, and new capillary vessels were formed [7]. In addition to these vascular effects, changes in skin temperature (Tsk) following the application of NMES [6,8] were verified, although these were shown to be very variable, perhaps due to the wide variation in the frequencies or wave used.

In addition, although the majority of studies carried out in the field of rehabilitation have focused on analyzing the effects of different procedures at the local level, contralateral changes have also been noticed after applying acupuncture [9], external pneumatic compression [10], percutaneous electrolysis [11], or procedures to increase strength and coordination [12,13]. At the sensory threshold, they have also been recorded through the application of vibrotherapy [14]. These changes are explained by the phenomenon known as “cross-education”, which has also been achieved through the application of NMES [15,16].

The thermal effects resulting from NMES, as well as other strategies employed in the fields of rehabilitation and sports medicine, can be measured using infrared thermography (IRT), a precise and noninvasive technology [17]. Analysis of images captured by IRT enables the quantitative measurement of variations in the thermal pattern of surfaces under study, commonly called the “region of interest” (ROI), which are indicators of changes in metabolism, in hemodynamics, or in neuronal thermoregulatory processes [2,18]. Investigations that are based on the combination of thermal images with laser Doppler have postulated that Tsk can be translated into quantity of skin blood flow, so that at higher temperatures, there is greater skin blood flow, and vice versa [19].

The aim of this study is to determine whether unilateral application of NMES can result in local and cross-education thermal effects and to measure the duration of these effects when the current is interrupted. The influence of dominance on these thermal effects is also analyzed.

## 2. Materials and Methods

### 2.1. Experimental Approach to the Problem

The research described in the present paper is an experimental study involving four thermographic evaluations following NMES placement. The study was designed considering methodological recommendations described in key consensus documents in the field of thermography research [20], which emphasizes the importance of collecting information related to participants’ demographic, environment/camera setup, and recording/analysis.

### 2.2. Subjects

It was determined that the study should include at least 42 subjects (based on one-way ANOVA and with a two-arm dominant vs. nondominant design), based on a power level of 90% and 5% levels of significance. In order to accommodate a possible loss of 10% of subjects during the study, we increased the initial number to 47 subjects. In the end, two subjects were lost, so that 45 subjects were included in the final analysis, of which 25 were women and 20 were men. The general characteristics of the subjects were as follows: 22.11 ± 2.96 years old; 169.90 ± 8.36 cm height; and 23.91 ± 4.75 kg body mass. A subcutaneous fat measurement was performed to evaluate a possible influence of the local fat amount (thigh skin fold: 21.73 ± 8.31).

The subjects were apparently healthy university students aged between 18 and 26 years. The criteria for leaving the study included lack of compliance with instructions given prior to the commencement of thermographic evaluations. Exclusion criteria included the presence of acute or chronic thigh injuries and the presentation of any of the contraindications to the application of the current. A quick questionnaire was made to see if the subjects had had any type of illness with fever in the previous week or if they had any type of vascular disorder such as hypertension, prehypertension, and Raynaud’s syndrome. Any of these two reasons would be a reason for exclusion from the experiment. The health condition of each subject was assessed during enrollment, and performed by investigators or health professionals holding a minimum Master’s degree in health sciences and with proven clinical experience with patients. Conditions were assessed by questionnaire with reference to inclusion criteria and dismissing exclusion criteria.

All participants were informed of the procedure at a first session during which they also signed a document to confirm their informed consent to participate. All protocol procedures were approved by the Research Commission of the San Juan de Dios School of Nursing and Physical Therapy, Comillas Pontifical University, Madrid, supported within the framework research project C.P.-C.I.15/416-E. The ethical approval was obtained from the Hospital Clínico Universitario San Carlos on 1 September 2015. Participants were informed that they could leave the study by revoking their informed consent at any time. The research was carried out following the Declaration of Helsinki of 1975 [21].

### 2.3. Procedures

During the first session of the evaluation protocol, data regarding sex, age, motor dominance, weight, and height were recorded. The dominant leg was established as the leg that the subject supports first after a surprise posterior–anterior thrust. The subjects were given various instructions to be acted upon before the taking of measurements: (1) avoid the consumption of large meals, alcohol, coffee, and/or tobacco, the performance of massages in the thigh area, and the application of any type of cream, medication, and hot or cold gel, within 6 h before each measurement; (2) avoid doing physical activity in the 24 h previous to data collection.

The second session focused on the taking of the thermal images. This session was carried out in a room specially prepared for the occasion, always maintaining the existence of a soft and indirect light, and avoiding the exposure of participants to direct air flows. Using a thermo-hygrometer, temperature and relative humidity measurements were taken, obtaining values of 23.48 ± 0.84 °C and 35.04 ± 3.90%, respectively. Direct physical contact and excessive proximity to the participants were avoided. To take thermal images, the A Flir^®^ E60 thermographic camera, commonly used in research in the biomedical field, was selected due to its high optical resolution (320 × 240 pixels), thermal sensitivity (<0.05 °C), and good image frequency (60 Hz). This camera was placed so that the screen remained at all times fixed to a bar perpendicular to the ground, maintaining a distance of 160 cm from the lens to the anterior region of the subject’s thighs. We chose a Flir ^®^ Camera, versus any other [22], since these cameras are meant to provide superior picture quality and improved resolution. These types of medical devices are specifically designed for diagnostic use and they meet all European regulations. In addition, the camera used was regularly inspected, checked, and controlled by the technician, and the facilities met the local regulations for health centers for clinical use [23,24]. We, therefore, consider thermography a reliable technique [25].

Prior to imaging, the placement of the NMES on the dominant or nondominant side of the subject was randomized by simple and systematic aleatory numbers with a 1:1 distribution. This procedure was performed only on the first subject of the day, and the placement then altered between the dominant and nondominant sides of the remaining subjects

When the images were taken, the subjects, wearing shorts, lay down in a supine position on a mattress covered with an opaque background material and kept their legs the same distance apart, as determined by a 15 cm wide expanded polystyrene mold.

In order to achieve the most effective stimulation of the quadriceps muscle, two small electrodes (5 × 5 cm) were placed on the distal points of both vastus muscles; approximately 2 cm from the upper edge of the kneecap in the area of the internal vastus and 5 cm from the upper edge of the kneecap in the external vastus. A third electrode (5 × 10 cm), with two outputs that closed both channels (internal vastus and external vastus), was located in the proximal and anterior third of the anterior rectum, 5 cm below the anterior superior iliac spine (Figure 1). This positioning was maintained for the duration of the shots included in the research protocol.

After placement of the electrodes, and prior to the start of the imaging protocol, each subject was given a standardized acclimation time of 10 min.

Once the first image was taken (Pre), the selected NMES pattern began (symmetric biphasic square, 400 μs, 8 Hz, 12 min, at the maximum intensity tolerated by the subject, applying a final average intensity of 27.3 ± 3.2 mA). Immediately after the current was turned off, with the electrodes still in place, a second thermal image was taken of the anterior region of the thigh (Post0). After an interval of 10 min, a third thermal photograph was taken (Post10). After a further 10 min of rest, a final photograph was taken, 20 min after the end of the NMES (Post20). The same exact areas were photographed and analyzed at different times for each participant.

#### Thermographic Analysis

All thermal images were analyzed using the FLIR tools plus software. Using this program, the ROI was delimited by the muscles on which the electrostimulation was applied (external and internal vastus and the anterior rectum of the quadriceps), leaving the electrodes used outside the said area. (Figure 1). For the region analyzed, we considered that the maximal Tsk was within the region of interest for all measurements.

The analyzed temperatures and the selected regions were compared with the contralateral leg, in which the selection of the ROI followed the same procedure.

### 2.4. Statistical Analyses

Only the subjects who successfully completed the two sessions were included in the statistical analysis, which was performed using IBM SPSS (IBM Corp. Released 2015. IBM SPSS Statistics for Windows, Version 23.0. Armonk, NY, USA: IBM Corp).

Once the normal distribution of the sample was determined (Kolmogorov–Smirnov test) and homogeneous behavior confirmed, evidenced by Mauchly′s sphericity test, a one-way ANOVA was performed for repeated measurements. The ANOVA was carried out with the objective of determining the effect of the independent variable “time” on the dependent variable “temperature” both in the leg where NMES was applied, and in the nonstimulated leg. In addition, the same analysis was performed for subjects in whom NMES was applied to the dominant leg on the one hand and for subjects in whom NMES was applied to the nondominant leg on the other. Where significant differences were found, a Bonferroni-adjusted post hoc test was performed to determine in more detail the specific differences between each of the time measurements found. The significance level was set at *p* ≤ 0.05.

Data are reported as mean and standard deviation, and reported *p*-values are those following Bonferroni correction.

In addition, a linear regression was performed to determine the cross-education between the legs (with and without NMES) in order to explore if there were differences in dominance. It was considered at a significance level of α = 5%.

Similarly, the possible existence of significant differences regarding the maximal temperature and sex of the subjects was also analyzed, as well as a possible correlation between the fat fold of the thigh and the temperature observed.

## 3. Results

We only obtained complete data from 45 out of 47 subjects. One of the subjects had taken coffee in the hour prior to the intervention, and another of them did not show up for the session.

The results of the subcutaneous fat measurement did not provide a significant correlation with the temperature increase.

Statistically significant differences were found in the maximal Tsk, both in the anterior part of the thigh where the NMES was performed, and in terms of cross-education in each of the four measurement periods (Pre, Post0, Post10, and Post20) with an F (3, 0.100) = 7.654; *p* < 0.000 in the electro-stimulated thigh, and F (3, 0.082) = 4.153; *p* = 0.008 in the contralateral.

After the post hoc analysis, adjusted by the Bonferroni test, statistically significant differences were found, showing an increase in temperature in all the measurements taken compared with the Pre measurements in the leg subjected to NMES (Post0, *p* = 0.001; Post10, *p* = 0.006; Post20, *p* = 0.003) (Figure 2A and Table 1).

In the nonstimulated leg, we also found an increase in temperature in all the measurements compared with the Pre measurements (Post0, *p* = 0.007; Post10, *p* = 0.043; Post20, *p* = 0.014) (Figure 2A).

There were no significant differences in the leg subjected to NMES between the Post0 and Post10 measurements (*p* = 0.98), between the Post0 and Post20 measurements (*p* = 1), or between the Post10 and Post20 measurements (*p* = 1). The same was observed in the leg which was not submitted to NMES between the Post0 and Post10 measurements (*p* = 0.1), between the Post0 and Post20 measurements (*p* = 1), and between the Post10 and Post20 measurements (*p* = 1).

When the NMES was applied to the dominant leg, the increase in temperature was 0.79% (*p* = 0.007) between the Pre and Post0 measurements, and 1.01% (*p* = 0.001) between the Pre and Post20 measurements. In the leg that did not receive NMES, the temperature increase was 0.55% (*p* = 0.037) and 0.4% (*p* = 0.09), respectively, between the Pre and Post0 measurements and the Pre and Post20 measurements (Figure 2B).

When the NMES was placed on the nondominant side, the temperature increase in the leg subjected to NMES was 0.53% (*p* = 0.318) between the Pre and Post0 measurements and 0.49% (*p* = 0.538) between Pre and Post20 measurements. In the leg that did not receive NMES, the temperature increase was 0.55% (*p* = 0.146) and 0.41% (*p* = 0.539), respectively, between the Pre and Post0 measurements and the Pre and Post20 measurements (Figure 2C and Table 1).

The results of the maximum average temperatures, the percentage variations in the temperature between each measurement, and the Bonferroni-adjusted *p*-values are shown in Table 1.

Linear regression analysis showed the existence of a positive relationship between the temperatures of the legs with and without NMES both for the Post0 measurements and for the Post10 and Post20 measurements, which is explained by the equations; *y* = 5.179 + 0.855*x*, *y* = 2.807 + 0.923*x*, and *y* = 2.807 + 0.923*x*, respectively (Figure 3).

Table 2 shows the results of the linear regression: the typified coefficients and their probability values for the three measurements, which make it possible to explain a causal relationship between the increase in temperature of the leg not receiving NMES with the application of the method on the contralateral leg.

Finally, regarding the correlations obtained with respect to the relationship between temperature increase and sex, no statistically significant results were found, as was the case between the results of the measurement of the skin fold of the thigh and the increase in temperature.

### 3.1. Skin Temperature Increase in the Leg With NMES

As a first result, we found an increase in the Tsk of the homolateral leg after the application of NMES (8 Hz, 400 μs width of pulse), which was evidenced immediately (0.67%), at 10 min (0.14%), and remained for 20 min, after which it fell by 0.05%.

The temperature increase observed in our study (0.35 °C) is lower than the one obtained by Toro and Poy [8] with a current of 100 Hz (+3.7 °C), perhaps due to the fact that the study measured the temperature just below the electrodes, thus increasing it. On the other hand, Novotny et al. [26] showed an increase in temperature of 0.04°C after voluntary exercise. The fact that a contraction induced by NMES achieves a greater increase in blood flow than a voluntary contraction has already been found in previous studies [7,27].

The current parameter used in our case (8 Hz) coincides with that used by several authors [7,28]. Other authors such as Araújo et al. [29] applied higher frequencies (50 Hz), accounting for the increase of blood flow as a result of the stimulation of type I muscle fibers. Furthermore, these authors also showed an increase in the number of blood capillaries, which rose to their highest point over a period of 15 days during the 10 min that the current was being applied [29].

Regarding the vascular effects that continued after the application of NMES, Petrofsky et al. [30] reported a 12% increase in blood flow that lasted for at least 5 min after the cessation of the current. By contrast, Varatharajan et al. [31] reported the disappearance of any effects after the completion of the NMES. It should be remembered that the pulse durations applied in this investigation reached 940 μs, much higher than that used in our work or used by other authors, who used currents that ranged between 70 and 400 μs. On the other hand, Boerner et al. [6] determined that the temperature increase after the application of an infrared therapy, followed by a diadynamic current, was maintained for at least 30 min after the application of the therapy.

### 3.2. Cross-Education

Another interesting finding of our work was the thermal increase produced in the contralateral leg after the immediate application of the current (0.55%), which lasted for up to 10 min.

Through Doppler laser, Ye and Griffin [32] reported vascular effects of cross-education related to the application of mechanical vibrations. Other studies, such as that of Doi et al. [14], show changes of temperature, although not significant, measured with a thermometer after the application of ipsilateral and contralateral vibrotherapy.

The characteristics of cross-education following NMES demonstrated in our study are in line with the results obtained in other works [13,15,16]. In the present study, we found an increase of 0.55% in the temperature of the unworked leg, while Bezerra et al. [15] found an increase in strength of 28% in the unworked leg. The NMES presents greater muscle activation as measured through surface electromyography [4]. In addition, there are some supraspinal adaptations derived from an activation of sensory and motor areas in the cortex, which are assumed following studies that demonstrated the phenomenon of cross-education in the increase of maximum voluntary contraction [16]. Along the same lines, Han et al. [33] demonstrated an increase in the activation of the primary sensory–motor area of the contralateral cortex on the side where the NMES was placed, and a bilateral activation of the supplementary motor areas. Blickenstorfer et al. [34], using functional magnetic resonance, demonstrated an activation of the contralateral primary motor cortex, premotor cortex, and cerebellum.Although most studies have investigated the phenomenon of cross-education in the area of strength, our research shows a cross-education effect on Tsk. Escamilla-Galindo et al. [12] and Manca et al. [35] state that the causes of the increase in temperature in the leg that is not exercised are not clear, since other parameters such as phospho-creatinekinase levels or electromyography (EMG), were not measured in their study in parallel with temperature. 

The phenomenon of temperature cross-education suggests a direction for future research that could consider, in a more detailed way, new zones of the cortex that could be stimulated by NMES, in addition to the primary motor cortex that controls strength.

### 3.3. Dominance

Another interesting result was to verify that the application of the NMES on the dominant side achieved a higher temperature cross-education effect than when applied on the nondominant side.

Regarding this important aspect, it should be highlighted that our results agree with the study conducted by Hortobágyi and Maffiuletti [16], who reported a greater transfer of temperature, due to cortical mechanisms whose precise role is still uncertain, when the limb on which the NMES was applied was the dominant one, compared with when it was applied to the nondominant one.

Some authors have even found an absence of significant cross-education results when strength training was performed on the nondominant side [12], results that are in line with ours regarding nonsignificant temperature increases when the NMES was applied on the nondominant side.

### 3.4. Study Limitations

Among the limitations of this research, it is necessary to mention the stabilization period used (10 min), which could have led to a higher Tsk in our subjects prior to the placement of the current than in other studies. We also believe it would have been appropriate to extend the completion of the last thermographic evaluation (Post20) to at least 30 min. Moreover, the increase in local temperature produced by NMES cannot be separated from a possible increase in temperature produced by a stimulation of the sympathetic nervous system. Another limitation of the study is the fact that the time of the menstrual cycle of the participating women, in which temperature variations can occur, was not considered.

Another limitation regarding food control over subjects could be the limited standardized window around the taking of a simple meal, but this is very different to nutritional control or meal composition control to limit the impact of foodstuffs (or lack thereof) on outcomes. It is, however, true that there was a control over the consumption of large meals. 

In our clinical practice, it should be noted that the effects obtained by using an electric current varied in duration depending on the side where the current was applied. Thus, any increase in blood flow produced by the current will reinforce the application of subsequent techniques that benefit from an increase in prior blood flow, such as iontophoresis, sonophoresis, or manual techniques, but will do so for 20 min if applied on the dominant side, and only for 10 min if applied on the nondominant side. These research outputs, which should be further investigated, are in line with the findings regarding the kinesio taping technique by Lemos et al. [36], where the results obtained varied according to the level of dominance of the bandaged member.

Finally, our study does not show significant differences between the maximal Tsk and the sex of the participants, which differs from the results found by Maffiuletti et al. [37] and Boerner et al. [6]. We believe that our results may be due to the fact that we worked at the maximum current intensity tolerated by the patient, which meant that the difference in impedance due to the thickness of the fat layer of each sex-dependent subject was corrected by a greater increase in the current intensity used.

We also consider that more studies are needed to affirm that the increase in temperature achieved can generate physiological or clinical changes, both locally and contralaterally.

### 3.5. Practical Applications

Our results may be useful and should be taken into account when treating various injuries; for example, in reactive phases of tendinopathies, an application of this type of current would generate an increase in the blood flow of the area, something normally not desired for this phase of the process but that would be very useful in the case of a degenerative phase or deterioration. In addition, although NMES is not the rehabilitation technique of choice for increasing tissue temperature, it might be useful in cases where techniques such as radiofrequency are contraindicated.

In cases where the current cannot be applied locally due to contraindications (e.g., plaster, wounds, and burns), an improvement in blood flow to the contralateral leg can help alleviate possible adverse vascular effects resulting from prolonged immobilization or in cases of local deterioration in circulation, whether due to circulatory or nervous system issues. Further, in the case of burns or wounds, where the application of other types of devices that promote an increase in blood flow is contraindicated, the application of electrical currents to the healthy contralateral leg can increase the blood flow in the injured leg.

It is also important to be aware that the application of NMES in the treatment of a limb may cause effects in the contralateral limb that may or may not be desired.

## 4. Conclusions

Based on the results of our study, it is possible to conclude that the application of a symmetrical biphasic current, with a frequency of 8 Hz and a pulse duration of 400 μs, applied for 12 min to the anterior region of the thigh, causes an increase in temperature of 0.35 °C locally, that is, over the same area to which the NMES is applied, which is maintained for at least 20 min after the use of the current.

Similarly, the application of NMES with these parameters in one leg produces an increase in temperature in the contralateral leg; although this does not extend beyond 10 min after the application of the current, it is greater when the NMES is applied on the dominant side.

Our study, which focuses on the analysis of the maximal Tsk, provides the specific current parameters that generate a local increase, as well as knowledge about how to generate a temperature increase on the contralateral side, which is very useful when the technique cannot be used at the local level due to medical contraindication. This has useful applications in clinical practice. In addition, this study suggests that future research could be usefully carried out into the zones of the cortex stimulated by NMES: if we know that the primary cortex is stimulated when the phenomena of cross-education in strength occurs, we should be able to find out what happens in relation to cross-education effects on temperature.

## Figures and Tables

**Figure 1 ijerph-17-09028-f001:**
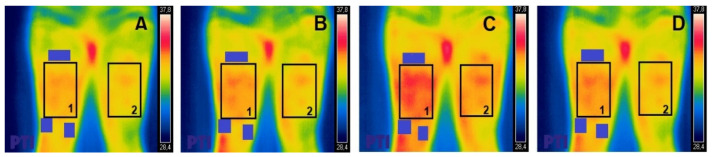
Thermal photography of the anterior region of the thigh (bilateral). Region of interest (ROI) indicated as 1 (leg with electrodes) and 2 (leg without electrodes); electrodes outside the ROI indicated by purple rectangles. (**A**) = Pre; (**B**) = Post0; (**C**) = Post10; (**D**) = Post20.

**Figure 2 ijerph-17-09028-f002:**
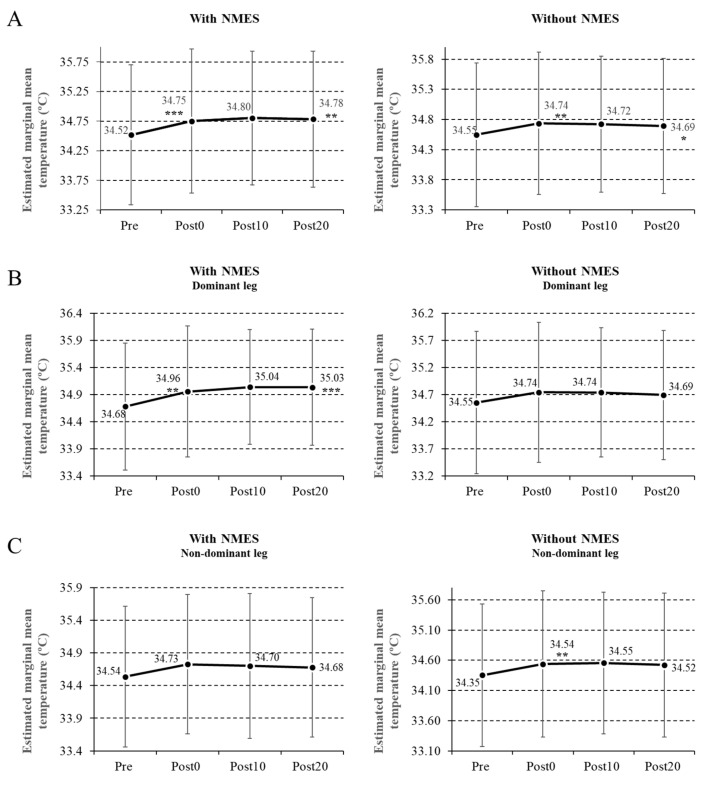
Variations of maximum temperature in the leg subjected to neuromuscular electrostimulation (NMES) and in the contralateral leg (**A**); temperature variations in both legs when the NMES was applied in the dominant leg (**B**); temperature variations in both legs when the NMES was applied in the nondominant leg (**C**). Data are reported as mean ± SD (error bars), and *p*-values reported are those following Bonferroni correction (* *p* < 0.05; ** *p* < 0.01; *** *p* < 0.001).

**Figure 3 ijerph-17-09028-f003:**
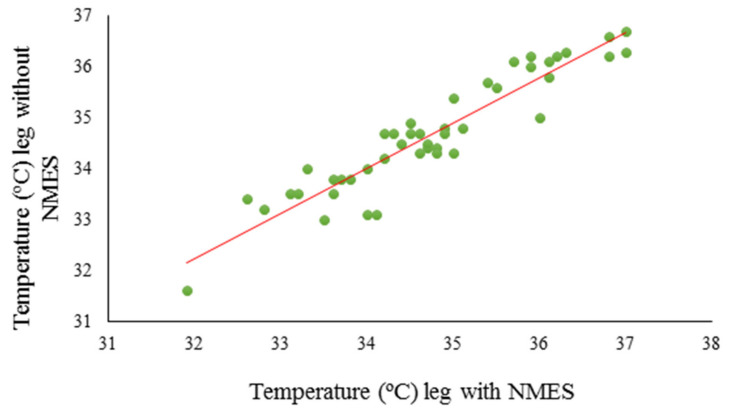
The temperature value of the leg that does not receive NMES can be predicted using the temperature value of the leg with NMES.

**Table 1 ijerph-17-09028-t001:** Temperature from each of the measurements (mean ± SD) and temperature change between one and other measures (difference (Diff) Pre–Post0; Diff Post0–Post10; Diff Post10–Post20; Diff Pre–Post20).

Figure		Pre	Post0	Post10	Post20	Diff Pre–Post0	Diff Post0–Post10	Diff Post10–Post20	Diff Pre–Post20
2A	E	34.52 ± 1.18	34.75 ± 1.22	34.80 ± 1.13	34.78 ± 1.15	0.233 ***	0.051	−0.018	0.266 **
	No E	34.55 ± 1.19	34.73 ± 1.18	34.72 ± 1.13	34.69 ± 1.12	0.189 **	−0.037	−0.029	0.147 *
2B	Dominant E	34.68 ± 1.17	34.96 ± 1.21	35.04 ± 1.06	35.03 ± 1.07	0.274 **	0.083	−0.005	0.352 ***
	Nondominant No E	34.55 ± 1.31	34.74 ± 1.29	34.74 ± 1.19	34.69 ± 1.19	0.191	−0.002	−0.048	0.141
2C	Nondominant E	34.35 ± 1.18	34.54 ± 1.21	34.55 ± 1.17	34.52 ± 1.19	0.182 **	0.018	−0.002	0.168
	Dominant No E	34.54 ± 1.08	34.73 ± 1.07	34.70 ± 1.11	34.68 ± 1.07	0.191	−0.024	−0.023	0.144

Figure 2A–C, relate the data with Figure 2. Dominant E—dominant leg with NMES; Nondominant No E—nondominant leg without NMES; Nondominant E—nondominant leg with NMES; Dominant No E—dominant leg with NMES; E—leg with NMES independently of the dominance; No E—leg without NMES independently of the dominance. Bonferroni-adjusted *p*-values (* *p* < 0.05; ** *p* < 0.01; *** *p* < 0.001).

**Table 2 ijerph-17-09028-t002:** Typified coefficients (B) and their probability values for the application moments of NMES (Post0, Post10, and Post20).

		B ± SD	Standardized B
Post0	Constant	5.179 ± 1.455	0.977 **
	Maximum temperature in the leg with NMES	0.855 ± 0.042	0.954 ***
Post10	Constant	2.807 ± 1.561	0.977 ***
	Maximum temperature in the leg with NMES	0.923 ± 0.450	0.954
Post20	Constant	4.563 ± 1.641	0.972 ***
	Maximum temperature in the leg with NMES	0.872 ± 0.480	0.944

** *p* < 0.01; *** *p* < 0.001.

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
