# Peer review of "Local and Contralateral Effects after the Application of Neuromuscular Electrostimulation in Lower Limbs"

_ijerph, 2020, doi:10.3390/ijerph17239028_

Round 1
Reviewer 1 Report
I read with interest the article by Benito-Martinez et al regarding the effects of NMES on skin temperature/blood flow. While the article was of interest, I do have several concerns which require address/response prior to any reconsideration and review of the discussion. Below are my comments:
Page 1, line 33: remove “In the literature”
Page 1, line 36: suggest “..some studies have shown increases in…”
Page 1, line 41: suggest “..which have been shown to be…”
Page 2, line 43: suggest “On the other hand,…”
Page 2, lines 44-47: See also https://doi.org/10.7717/peerj.4878
Page 2, lines 55-58: I believe this statement is referring to cutaneous/skin blood flow. Please address.
Page 2, lines 74-75: (1) suggest “…and 67.78 ± 12.00 kg body mass.”; (2) Please report BMI here; (3) Was there an assessment of subcutaneous fat locally/regionally or total body?
Page 2, line 76: Define healthy. Were these apparently healthy subjects? Was a health questionnaire administered? Was blood pressure measured at baseline (i.e., devoid of pre-hypertension or hypertensive status)? Were any of the subjects taking medications? Vasoactive substances? Since women were included in the study, was the effect of the menstrual cycle on cutaneous blood flow patterns or the effect of the intervention considered?
Page 2, line 76-77: suggest revision of wording to assure that participation in the study was voluntary and participants could leave the study at any time. When reporting on research that involves human subjects, human material, human tissues, or human data, authors must declare that the investigations were carried out following the rules of the Declaration of Helsinki of 1975 (https://www.wma.net/what-we-do/medical-ethics/declaration-of-helsinki/), revised in 2013.
Page 2, line 87: suggest ‘sex’ in place of ‘gender’.
Page 2, line 97: has this camera been validated as a tool for infrared thermography?
Page 2, line 101: Was the time between laying down and the first image being taken standardized? Or the time after placement of the electrode and the first image being taken standardized? Essentially, was there sufficient and standardized time for equilibration to environmental conditions prior to imaging?
Page 3, lines 90-91: Was there any standardization for foodstuff consumption beyond alcohol/coffee? GI blood flow being variable between visits/participants could be a significant confounding factor.
Page 4, line 149: suggest “Bonferroni adjusted”
Page 4, lines 151-152: (1) How?; (2) suggest sex instead of gender; (3) lines 217-218 suggest this was explored via correlation? Would this have not been better to look at as a covariate or additional factor?
Page 4, lines 167-: Are the reported p-values the Bonferroni adjusted values? If not, please report in methods how the family wise error rate was adjusted. For example, if looking at only with NMES in panel A, as reported I believe that 5 comparisons were made. Thus, a p-value of <0.01 would be required for significance ([alpha]0.05/[# of comparisons]5).
Page 5, figure 2. (1) The error bars for post10 time points are out of scale for panel a & b. Please adjust scale accordingly. (2) The caption needs to include information with regards to the symbols on the figure and what they represent (*, **, etc.). Are the data reported as mean ± SD or sem? This should be included here as well as in the methods of the text.
Page 5, figure 2: Were the baseline temperature between legs and between the conditions compared? For example, the difference in temperature at baseline in panels b & c for with NMES is concerning.
Page 5, lines 174-175: Do these p-values meet criteria for significance with Bonferroni correction? See earlier comment(s). Comparisons were made between all time points.
Page 5, lines 177-181. Please again refer to panel A as it is confusing which set of data are being referred to
Results: Please consider actual effect sizes (degrees change) in addition to or as opposed to percent change as this may be more informative for the observed changes in skin temperature
Page 10, line 308: What is the clinical or practical effect of the increase of temperature observed herein? Does it equate to a relative amount of limb perfusion? Does the temperature change meet the threshold for a significant clinical difference?
Author Response
Answers to Reviewer 1
We would like to thank for the reviewer effort and dedicated time to evaluate our manuscript. The points addressed by the reviewer allowed us to improve considerably our work.
I read with interest the article by Benito-Martinez et al regarding the effects of NMES on skin temperature/blood flow. While the article was of interest, I do have several concerns which require address/response prior to any reconsideration and review of the discussion. Below are my comments:
Page 1, line 33: remove “In the literature”
Removed, please see the revised version of the manuscript, line 31.
Page 1, line 36: suggest “..some studies have shown increases in…”
Suggestion accepted and included in the revised version of the manuscript, line 34.
Page 1, line 41: suggest “..which have been shown to be…”
Suggestion accepted and included in the revised version of the manuscript, line 39.
Page 2, line 43: suggest “On the other hand,…”
Suggestion accepted and included in the revised version of the manuscript, line 41.
Page 2, lines 44-47: See also https://doi.org/10.7717/peerj.4878
Thank you for the suggestion. The work by Martin and coauthors has been added to the revised version of the manuscript, line 43. The full reference was added in the References section, lines 418-420.
Page 2, lines 55-58: I believe this statement is referring to cutaneous/skin blood flow. Please address.
In order to clarify this particular part of the manuscript, we added the word “skin” where necessary. Please see the revised version of the manuscript, line 55.
Page 2, lines 74-75: (1) suggest “…and 67.78 ± 12.00 kg body mass.”; (2) Please report BMI here; (3) Was there an assessment of subcutaneous fat locally/regionally or total body?
In order to address the reviewer concerns we carried out a subcutaneous fat measurement. However, there was no correlation with the temperature increase. New elements were added in the revised version of the manuscript. Please see lines 74-76.
Page 2, line 76: Define healthy. Were these apparently healthy subjects? Was a health questionnaire administered? Was blood pressure measured at baseline (i.e., devoid of pre-hypertension or hypertensive status)? Were any of the subjects taking medications? Vasoactive substances? Since women were included in the study, was the effect of the menstrual cycle on cutaneous blood flow patterns or the effect of the intervention considered?
To all participants was applied a quick questionnaire was made to see if the subjects had had any type of illness with fever in the previous week or if they had any type of vascular disorder. Any of these two reasons would be a reason for exclusion from the experiment. This information was added to the revised version of the manuscript, lines 80-83.
Regarding the menstrual cycle of the participating women, it has not been considered. Even though all the temperature measurements were taken on the same day, and the differences were assessed, we do not believe that it played a crucial role in the results. Despite this we add this information to the Study limitations section, lines 331-333
Page 2, line 76-77: suggest revision of wording to assure that participation in the study was voluntary and participants could leave the study at any time. When reporting on research that involves human subjects, human material, human tissues, or human data, authors must declare that the investigations were carried out following the rules of the Declaration of Helsinki of 1975 (https://www.wma.net/what-we-do/medical-ethics/declaration-of-helsinki/), revised in 2013.
The ethical aspects are of major importance, thank you. We added information regarding the voluntary participation, and we assure that the guidelines of the Declaration of Helsinki of 1975 were followed. This information was included in the revised version of the manuscript, lines 89-91.
Page 2, line 87: suggest ‘sex’ in place of ‘gender’.
Suggestion accepted and included in the revised version of the manuscript. We changed every “sex” entry by “gender”.
Page 2, line 97: has this camera been validated as a tool for infrared thermography?
Yes, the Flir® E60 camera refers to a high resolution camera and is validated as an infrared thermography instrument based on its high optical resolution (320 × 240 pixels), thermal sensitivity (< 0.05ºC) and good image frequency (60 Hz). FLIR type infrared cameras are commonly used in the detections of biomedical problems, since they have focal plane array detectors, which work with thermal detectors. In order to provide this information to the reader, we added this information in the revised version of the manuscript, lines 107-109.
Page 2, line 101: Was the time between laying down and the first image being taken standardized? Or the time after placement of the electrode and the first image being taken standardized? Essentially, was there sufficient and standardized time for equilibration to environmental conditions prior to imaging?
After placement of the electrodes, and prior to the start of the imaging protocol, each subject had a standardized acclimation time of 10 minutes. In order to clarify this aspect, this information was added to the revised version of the manuscript, lines 132-133.
Page 3, lines 90-91: Was there any standardization for foodstuff consumption beyond alcohol/coffee? GI blood flow being variable between visits/participants could be a significant confounding factor.
There was no standardization for food consumption beyond alcohol/coffee but, since thermographic evaluation appointments were scheduled in all cases in the middle of the morning (10 am to 1 pm). Imaging after lunch, which is the most abundant meal in the country where the study was conducted, was avoided. To clarify this, we added new information in the revised version of the manuscript, lines 98-101.
Page 4, line 149: suggest “Bonferroni adjusted”
Corrected in the revised version of the manuscript.
Page 4, lines 151-152: (1) How?; (2) suggest sex instead of gender; (3) lines 217-218 suggest this was explored via correlation? Would this have not been better to look at as a covariate or additional factor?
An additional factor was analyzed in a previous pilot study related to skin color. In that previous study we did not found any significant differences. In this study, it was not our objective
Page 4, lines 167-: Are the reported p-values the Bonferroni adjusted values? If not, please report in methods how the family wise error rate was adjusted. For example, if looking at only with NMES in panel A, as reported I believe that 5 comparisons were made. Thus, a p-value of <0.01 would be required for significance ([alpha]0.05/[# of comparisons]5).
If you refer to the Bonferroni values that are shown in Table 1, this information is clarified by modifying Table 1 and with the text of lines 219-221.
Page 5, figure 2. (1) The error bars for post10 time points are out of scale for panel a & b. Please adjust scale accordingly. (2) The caption needs to include information with regards to the symbols on the figure and what they represent (*, **, etc.). Are the data reported as mean ± SD or sem? This should be included here as well as in the methods of the text.
Figure 2 has been revised according to the reviewer suggestions. Please see the revised version of the manuscript, line 196.
Page 5, figure 2: Were the baseline temperature between legs and between the conditions compared? For example, the difference in temperature at baseline in panels b & c for with NMES is concerning.
Figure 2 has been revised according to the reviewer suggestions. Please see the revised version of the manuscript, line 196.
Page 5, lines 174-175: Do these p-values meet criteria for significance with Bonferroni correction? See earlier comment(s). Comparisons were made between all time points.
Yes, they meet the criteria. Clarifications were made on Table 1 and Figure 2.
Page 5, lines 177-181. Please again refer to panel A as it is confusing which set of data are being referred to
To clarify the data, Table 1 and Figure 2 were modified.
Results: Please consider actual effect sizes (degrees change) in addition to or as opposed to percent change as this may be more informative for the observed changes in skin temperature
We changed the percentage by degree change.
Page 10, line 308: What is the clinical or practical effect of the increase of temperature observed herein? Does it equate to a relative amount of limb perfusion? Does the temperature change meet the threshold for a significant clinical difference?
We do not know if these temperature changes can generate clinical changes or not, but it seems to us a very important appreciation. We included a sentence that recommends further studies to be able to confirm this in the revised version of the manuscript, line 348-349.

Reviewer 2 Report
The research reported in this manuscript investigated the effect of unilateral NMES on skin temperature changes of both sides. The effects of stimulating the dominant and non-dominant leg were also compared. It was concluded that there was a cross-education effect on skin temperature and application of NMES on the dominant leg resulted in greater effect.
The study produced new and interesting findings in respect of the effects of unilateral NMES, if the authors could clarify the following points.
Line 70: please clarify whether the estimation of the minimum number of participants were based on One-Way ANOVA (or Two-Way ANOVA) with (or without) a cross over design. If there was no cross over, you had 22 and 23 participants in dominant leg and non-dominant leg stimulation groups respectively. But if there was a cross over, there would be 45 under each condition.
Line 87-91: should also control for physical activities that would certainly influence blood flow through the limb.
How was the leg dominance determined?
Line 109-112: how was the motor points determined?
How did you make sure the skin temperature was stabilised prior to thermal imaging, e.g. how long the participants rested and whether you had taken multiple measurements at set time intervals to ensure the skin temperature was stable (or within an acceptable range).
Line 116: please spell out ROI for the first time use, region of interest?
Line 120: what was the range of the current or voltage. The max tolerance level could be quite different between participants.
Line 126-130: please clearly state whether the exactly same areas were analysed at different time points for each participant.
Line 144-150: the description of ANOVA is confusing. There was one independent variable “time” and you measured one dependent variable “temperature”, ie. you had only obtained the temperature data at four time points. So it was not to test “time x temperature” effect (that would be a two-way ANOVA).
In fact, if you were to test whether the responses of the two legs were different, then you should use a two-way ANOVA for the effect and interaction of “leg x time” (two independent variables) for the temperature (one dependent variable) change.
How was the difference between genders tested? Where to find the results? Was the potential effect of menstrual cycle on temperature considered (or should not be considered) in this study?
Line 170-173: the caption for Fig.2 could be made clearer. Was the panel (A) presenting the combined results regardless of leg dominance (ie. 45 for each leg)? And in panel (B) and (C) the number would be 22 for NMES on dominant leg, and 23 on non-dominant leg?
Line 174: consider use “In the non-stimulated leg”.
Line 208-211: I don’t understand this table. The statistical method was not described in Methods section.
Line 302-307: authors might consider summarise all limitations of the study in one paragraph.
Line 133-139, 215-216: it is not clear under what conditions the EMG was recorded, e.g. was the participant instructed to fully relax the muscles?
Line 230-231: the discussion on acute and chronic effects of NMES should not be mixed here.
Line 245-289, 338-340: although this study investigated the acute effect of unilateral NMES on the skin temperature of both legs, the discussion should not be mixed with the cross-education effects of chronic stimulation (training) on muscle strength, and the regulatory mechanisms might be different as the subcutaneous blood flow is regulated via the autonomic nervous system.
Line 290-293: the stabilisation period was not introduced in Methods.
Author Response
Answers to Reviewer 2
We would like to thank for the reviewer effort and dedicated time to evaluate our manuscript. The points addressed by the reviewer allowed us to improve considerably our work.
The research reported in this manuscript investigated the effect of unilateral NMES on skin temperature changes of both sides. The effects of stimulating the dominant and non-dominant leg were also compared. It was concluded that there was a cross-education effect on skin temperature and application of NMES on the dominant leg resulted in greater effect.
The study produced new and interesting findings in respect of the effects of unilateral NMES, if the authors could clarify the following points.
Line 70: please clarify whether the estimation of the minimum number of participants were based on One-Way ANOVA (or Two-Way ANOVA) with (or without) a cross over design. If there was no cross over, you had 22 and 23 participants in dominant leg and non-dominant leg stimulation groups respectively. But if there was a cross over, there would be 45 under each condition.
This aspect has been clarified in the revised version of the manuscript, please see lines 68-69.
Line 87-91: should also control for physical activities that would certainly influence blood flow through the limb.
Yes, it has been taken into account that the participants should not have practiced physical exercise in the 24 hours prior measurements. This information has been added in the revised version of the manuscript, lines 98-101..
How was the leg dominance determined?
New information added in the revised version of the manuscript related to this topic, please see lines 94-95.
Line 109-112: how was the motor points determined?
It has been eliminated. Actually, the electrodes were placed in anatomic points.
How did you make sure the skin temperature was stabilised prior to thermal imaging, e.g. how long the participants rested and whether you had taken multiple measurements at set time intervals to ensure the skin temperature was stable (or within an acceptable range).
This is a pertinent comment, thank you. After placement of the electrodes, and prior to the start of the imaging protocol, each subject had a standardized acclimation time of 10 minutes. In order to clarify this aspect, this information was added to the revised version of the manuscript, lines 132-133.
Line 116: please spell out ROI for the first time use, region of interest?
The first entry of “ROI” is on line 52. The full description has been included in the original version of the manuscript.
Line 120: what was the range of the current or voltage. The max tolerance level could be quite different between participants.
The maximum intensity tolerated by the subject is an intensity measurement used in most scientific investigations on electrostimulation since it allows a similar current dose to be applied to subjects regardless of the impedance generated by the fat fold. The average intensity data is added, as well as its standard deviation. Please see the revised version of the manuscript, lines 135-136.
Line 126-130: please clearly state whether the exactly same areas were analysed at different time points for each participant.
Information added to the revised version of the manuscript, lines 140-141.
Line 144-150: the description of ANOVA is confusing. There was one independent variable “time” and you measured one dependent variable “temperature”, ie. you had only obtained the temperature data at four time points. So it was not to test “time x temperature” effect (that would be a two-way ANOVA).
In fact, if you were to test whether the responses of the two legs were different, then you should use a two-way ANOVA for the effect and interaction of “leg x time” (two independent variables) for the temperature (one dependent variable) change.
Dear reviewer, that was not our goal. The explanation of the ANOVA has been changed so that it could be understood. Indeed, the time × temperature interaction has not been performed.
How was the difference between genders tested? Where to find the results? Was the potential effect of menstrual cycle on temperature considered (or should not be considered) in this study?
In the revised version of the manuscript, at the end of the results section it is detailed that no significant differences were found regarding gender (lines 249-250).
The potential effect of the menstrual cycle has not been taken into account in the article. However, we added this fact as a limitation of the study at section 4.4 Study limitations (lines 329-333).
Line 170-173: the caption for Fig.2 could be made clearer. Was the panel (A) presenting the combined results regardless of leg dominance (ie. 45 for each leg)? And in panel (B) and (C) the number would be 22 for NMES on dominant leg, and 23 on non-dominant leg?
Figure 2 has been revised according to the reviewer 1 suggestions. Please see the revised version of the manuscript, line 196.
Line 174: consider use “In the non-stimulated leg”.
Change included in the revised version of the manuscript, please see line 201.
Line 208-211: I don’t understand this table. The statistical method was not described in Methods section.
Table 1 has been considerably changed. Please see the new version on line 223.
Line 302-307: authors might consider summarise all limitations of the study in one paragraph.
In order to resume all study limitations, we created a new section, 4.4. Study limitations. Please see the revised version of the manuscript, lines 326-349.
Line 133-139, 215-216: it is not clear under what conditions the EMG was recorded, e.g. was the participant instructed to fully relax the muscles?
All information and data about the pilot study with EMG was removed, has also suggested by other reviewer.
Line 230-231: the discussion on acute and chronic effects of NMES should not be mixed here.
This particular part has been eliminated form the manuscript.
Line 245-289, 338-340: although this study investigated the acute effect of unilateral NMES on the skin temperature of both legs, the discussion should not be mixed with the cross-education effects of chronic stimulation (training) on muscle strength, and the regulatory mechanisms might be different as the subcutaneous blood flow is regulated via the autonomic nervous system.
In order to avoid causing confusion to reader, some information has been eliminated regarding this aspect.
Line 290-293: the stabilisation period was not introduced in Methods.
After placement of the electrodes, and prior to the start of the imaging protocol, each subject had a standardized acclimation time of 10 minutes. In order to clarify this aspect, this information was added to the revised version of the manuscript, lines 132-133.

Reviewer 3 Report
General: The paper theme is interesting with high relevance to clinical practice, consequently, to improve the quality of life to these subjects. The paper is well-written, just some major and minor points must be improved.
Major review
- L. 120: “at the maximum intensity tolerated by the subject”. This method may induce an error, indeed this intensity level is stressful to participants that triggers the sympathetic nervous system. In figure 1, the hip region shows a qualitative difference in temperature, I believe that the same occurs with all body. How can you split the real NMES effect on local temperature from the sympathetic nervous system? (Yes, I am the same reviewer of Sensor Journal). If you do not have an answer, include it with limitations in the discussion section.
- L. 139: The EMG parameters do not be described, as acquisition rate, filters, features, and others. In my opinion, the subtopic 2.3.2 can be removed (N=8), as your EMG results or insert in table 3 (N=8).
Minor review
- L. 163: change “P=0.000” to “P<0.001”. “P=0.000” do not exist, although the SPSS show in this format, the P value is smaller than the P=0.000, if you perform the same test in matlab or octave, you will check the real value.
- L. 170: boxplot must be used with non-parametric data.
- L. 222: change “impulse” to “pulse”.
Author Response
Answers to Reviewer 3
We would like to thank for the reviewer effort and dedicated time to evaluate our manuscript. The points addressed by the reviewer allowed us to improve considerably our work.
General: The paper theme is interesting with high relevance to clinical practice, consequently, to improve the quality of life to these subjects. The paper is well-written, just some major and minor points must be improved.
Major review
- 120: “at the maximum intensity tolerated by the subject”. This method may induce an error, indeed this intensity level is stressful to participants that triggers the sympathetic nervous system. In figure 1, the hip region shows a qualitative difference in temperature, I believe that the same occurs with all body. How can you split the real NMES effect on local temperature from the sympathetic nervous system? (Yes, I am the same reviewer of Sensor Journal). If you do not have an answer, include it with limitations in the discussion section.
The maximum tolerated intensity is currently used in most papers studying electrostimulation, any other defined intensity, would vary between subjects, the current density received depending on its impedance.
We consider that the qualitative temperature increase observed would not provide significant qualitative data, but it is true that they have not been studied. On the other hand, the stimulation of the sympathetic nervous system through electric current, we have not found that this is demonstrated with 8 Hz. For all these reasons, as suggested, we include this fact in limitations.
- 139: The EMG parameters do not be described, as acquisition rate, filters, features, and others. In my opinion, the subtopic 2.3.2 can be removed (N=8), as your EMG results or insert in table 3 (N=8).
All about EMG has been removed
Minor review
- 163: change “P=0.000” to “P<0.001”. “P=0.000” do not exist, although the SPSS show in this format, the P value is smaller than the P=0.000, if you perform the same test in matlab or octave, you will check the real value.
It has been changed in the revised version of the manuscript.
- 170: boxplot must be used with non-parametric data.
It has been changed in the revised version of the manuscript.
- 222: change “impulse” to “pulse”.
Correction made in the revised version of the manuscript.

Round 2
Reviewer 1 Report
The authors have significantly improved the manuscript as a result of the first revision. However, I still have the following suggestions that should be considered prior to further consideration for publication.
Lines 74-76. Please report BMI here. Also, please report the results of the subcutaneous fat measurements. Reports regarding the lack of a relationship between subcutaneous fat and observed effects should be reported in the results and/or discussion.
Line 77: Note as apparently healthy
Lines 80-83: Define vascular disorder. What does this include (e.g. hypertension, prehypertension, diabetes, etc.)
Lines 95-101: If the avoidance of physical activity was included with instructions, please combine into a single list with appropriate tense and conjunctions.
Lines 107-109: Include citation(s) for validation studies of the device for the purpose used herein
Lines 98-101: While a standardized window around a common meal can be appreciated, this is far different than nutritional control or meal composition control to limit the impact of foodstuffs (or lack thereof) on outcomes. Thus, I suggest adding a note in the limitations section of the manuscript with regards to this.
Line 177-178: How was this possible difference analyzed/measured?
Figure 2: The new figure is improved from the original. However, the removal of error bars and markers of significance has weakened the illustration. Suggest including those aspects as well as relevant information regarding the figure in the legend (e.g., how data is presented [mean +/- SD], what significance markers entail, etc.)
Line 220: Suggest, “..Bonferroni adjust p-values..”. Alternatively, suggest including a note in the statistical methods section that says data are reported as mean +/- SD and reported p-values are those following Bonferroni correction. Notably, the figure legends should include a notation of how the data are reported (i.e., mean +/- SD)
Table 1: Carefully check the values including and use appropriate number of significant digits. For example, mean and SD should have the same number of significant digits. Another example, the first value of the Diff Post0-Post10 column appears to be missing a decimal point
Overall, suggest going through manuscript for organization and flow of information. One example, the notations about exploration of effect of sex on observed outcomes feels isolated as opposed to being included in particular statistical models for consideration. Also, please consider citations in the limitations and practical applications sections of the manuscript to support statements and considerations
Author Response
Answers to Reviewer 1 remarks
We would like to thank for the reviewer effort and dedicated time to evaluate our manuscript. The points addressed by the reviewer allowed us to improve considerably our work.
Lines 74-76. Please report BMI here. Also, please report the results of the subcutaneous fat measurements. Reports regarding the lack of a relationship between subcutaneous fat and observed effects should be reported in the results and/or discussion.
In the revised version of the manuscript we addressed all your concerns. On line 77 we ended the sentence with the following information “thigh skin fold 21.73 ± 8.31”. Also, we address this topic in the material and methods (lines 183-185) and on the results section (lines 254-256).
Line 77: Note as apparently healthy
In the revised version of the manuscript we added the word “apparently” (line 78).
Lines 80-83: Define vascular disorder. What does this include (e.g. hypertension, prehypertension, diabetes, etc.)
In the revised version of the manuscript (lines 83-84) we included the main types of vascular disorders for the sample population under study: hypertension and prehypertension as well as Raynaud´s Syndrome due to affecting the cutaneous blood flow.
Lines 95-101: If the avoidance of physical activity was included with instructions, please combine into a single list with appropriate tense and conjunctions.
The mentioned lines were reformulated into a new paragraph to combine the measures in two points: those that should have been avoided 6 and 24 hours before taking any measurement. Please see the revised version of the manuscript (lines 103-107).
Lines 107-109: Include citation(s) for validation studies of the device for the purpose used herein
A new paragraph and three citations have been included in the text. Please see the revised version of the manuscript (lines 120-125).
Lines 98-101: While a standardized window around a common meal can be appreciated, this is far different than nutritional control or meal composition control to limit the impact of foodstuffs (or lack thereof) on outcomes. Thus, I suggest adding a note in the limitations section of the manuscript with regards to this.
To address this topic, a new paragraph has been included in the Limitations section. Please see the revised version of the manuscript (lines 329-332).
Line 177-178: How was this possible difference analyzed/measured?
A new sentence has been added in the same paragraph referred by the reviewer. Please see the revised version of the manuscript (lines 175-177).
Figure 2: The new figure is improved from the original. However, the removal of error bars and markers of significance has weakened the illustration. Suggest including those aspects as well as relevant information regarding the figure in the legend (e.g., how data is presented [mean +/- SD], what significance markers entail, etc.)
The error bars and markers of significance were added in the new figure 2 version.
Line 220: Suggest, “..Bonferroni adjust p-values..”. Alternatively, suggest including a note in the statistical methods section that says data are reported as mean +/- SD and reported p-values are those following Bonferroni correction. Notably, the figure legends should include a notation of how the data are reported (i.e., mean +/- SD)
The sentence on line 227 has been corrected to include “Bonferroni adjusted p-values” in the results section. Also, the sentence “Data are reported as mean and standard deviation and reported p-values are those following Bonferroni correction” in the section regarding statistical treatment (lines 178-179). The Figure caption was changed to address this aspect.
Table 1: Carefully check the values including and use appropriate number of significant digits. For example, mean and SD should have the same number of significant digits. Another example, the first value of the Diff Post0-Post10 column appears to be missing a decimal point
We adjusted Table 1 according to your comment but also following the Editor suggestion to match the number of decimal places between mean and SD. Please see Table 1 in the revised version of the manuscript.
Overall, suggest going through manuscript for organization and flow of information. One example, the notations about exploration of effect of sex on observed outcomes feels isolated as opposed to being included in particular statistical models for consideration. Also, please consider citations in the limitations and practical applications sections of the manuscript to support statements and considerations
We have considered the suggestions and rearranged the data for a better understanding.

Reviewer 2 Report
In the revised manuscript the authors have addressed most of my previous comments. Thank for their efforts. However, there are still some areas that require further clarification as listed below.
Line 69: the authors claimed that they used a cross over design, however in my understanding it was incorrect. For a cross over design, one half of the participants would receive NMES on one leg (e.g. dominant) in one session, and the other half of participants would receive NMES on the other leg (e.g. non dominant) in the same session. Then, in the next session, the participants will receive NMES on the other leg. However, in this study, each participant only received NMES in one leg (22 on dominant and 23 on non dominant), and there was no swap over session (ie. no cross over).
Figure 2 is confusing. There are two sections (bar chart and line graph) that appear to present the same mean values, but there is no explanation on why and how to interpret the data. It might be better to present the data in two figures if the authors believe it is necessary to present the data separately in two sections.
Table 2 is not cited in the manuscript (Line 237 is deleted). Also, it is not clear what are presented in the table. Are the coefficients for the relationship between the Tsk of dominant and nondominant legs?
The manuscript needs further editing for English presentation, particularly for the revised/added areas.
Author Response
Answers to Reviewer 2 remarks
We would like to thank for the reviewer effort and dedicated time to evaluate our manuscript. The points addressed by the reviewer allowed us to improve considerably our work.
Line 69: the authors claimed that they used a cross over design, however in my understanding it was incorrect. For a cross over design, one half of the participants would receive NMES on one leg (e.g. dominant) in one session, and the other half of participants would receive NMES on the other leg (e.g. non dominant) in the same session. Then, in the next session, the participants will receive NMES on the other leg. However, in this study, each participant only received NMES in one leg (22 on dominant and 23 on non dominant), and there was no swap over session (ie. no cross over).
Thank you for your correction, we understood a two-arm design as cross over, and we have substituted the conception as (Section 2.2. Subjects):
“It was determined that the study should include at least 42 subjects (based on one-way ANOVA and with a two-arm dominant vs. non-dominant design), based on a power level of 90% and 5% levels of significance.”, lines 70-72 of the revised version of the manuscript.
Figure 2 is confusing. There are two sections (bar chart and line graph) that appear to present the same mean values, but there is no explanation on why and how to interpret the data. It might be better to present the data in two figures if the authors believe it is necessary to present the data separately in two sections.
Figure 2 has been improved according to the suggestions of reviewer 1. Their interpretation has been clarified in the legend.
Table 2 is not cited in the manuscript (Line 237 is deleted). Also, it is not clear what are presented in the table. Are the coefficients for the relationship between the Tsk of dominant and nondominant legs?
We have included a paragraph that was in an earlier version and that was deleted by mistake. Please see the revised version of the manuscript, lines 245-246.
The manuscript needs further editing for English presentation, particularly for the revised/added areas.
Dear reviewer, the revised version of the manuscript has been proofread by a company. Please check the English editing certificate at readytopub.com with the following key: 6MTR3-07QY8-B1TKA-PA8HE-BP5ER

Reviewer 3 Report
Ok. My questions were answered.
Author Response
Answers to Reviewer 3 remarks
Ok. My questions were answered.
Dear reviewer, we would like to thank for the reviewer effort and dedicated time to evaluate our manuscript. Thank you for your help.
